# The Impact of Population Ageing on Rural Aged Care Needs in Australia: Identifying Projected Gaps in Service Provision by 2032

**DOI:** 10.3390/geriatrics8030047

**Published:** 2023-04-27

**Authors:** Irene Blackberry, Nicholas Morris

**Affiliations:** 1John Richards Centre for Rural Ageing Research, La Trobe Rural Health School, La Trobe University, Wodonga, VIC 3689, Australia; i.blackberry@latrobe.edu.au; 2Care Economy Research Institute, La Trobe University, Wodonga, VIC 3689, Australia; 3La Trobe Law School, La Trobe University, Melbourne, VIC 3086, Australia

**Keywords:** aged care, Australia, rural, ageing, regional, remote

## Abstract

This observational study examines and estimates the trends and impact of population ageing on rural aged care needs in Australia. With its universal health system and subsidised aged care system, Australia is among those countries with a long life expectancy. Being a geographically large country with a relatively small and dispersed population presents challenges for equitable access to aged care service provision. While this is widely acknowledged, there is little empirical evidence to demonstrate the magnitude and location of the aged care service provision gaps in the next decade. We performed time series analyses on administrative data from the Australian Bureau of Statistics and the Australian Institute of Health and Welfare GEN databases. The Aged Care Planning Regions (ACPR) were classified according to geographical remoteness using the Modified Monash Model scale. There is currently a shortfall of 2000+ places in residential aged care in rural and remote areas of Australia based on 2021 data. By 2032, population ageing will mean that an additional 3390 residential care places and around 3000 home care packages will be required in rural and remote communities alone. Geographical disparities in aged care exist in Australia and continue to worsen, requiring immediate action.

## 1. Introduction

The population is ageing as people live longer globally. Better public health provisions have contributed to increases in life expectancy for the past 200 years. Improvement in life expectancy does not only occur at birth but is seen across all ages. Geographical disparities exist, and this contributes to inequality in life expectancy between and within countries. Developed countries, such as Australia, have a life expectancy at birth that is over a decade higher than the global life expectancy of 73 years [1].

Recent studies have highlighted ongoing geographical disparities in equity of access to aged care services between urban and non-urban populations and the locational challenges they face [2,3]. Smaller population sizes and geographical isolation make traditional models of service provision problematic in regional and remote areas [4]. Older people in remote communities often still lack basic services and have insufficient economic and community resources. Factors such as distance, isolation, housing, income, access to services, and transport are significant for the positive and healthy ageing of rural older people [5,6].

While healthcare and aged care are often funded separately, there is a strong interface between the two systems. In Australia, most health and aged care services are subsidised by the government through separate funding streams and delivered by independent care providers. These services range from home care support for those who live independently at home to full-time care in a residential aged care setting. Over the years, there has been an increased preference for older people to live at home with support. A 2019 longitudinal study observed that as older Australians chose transition and home care packages, the number of those in residential care decreased [7]. The 2021–22 Report on the Operation of the Aged Care Act 1997 also found a one-fifth increase in the number of people utilising a home care package [8]. Previous reports provide information on the National Aboriginal and Torres Strait Islander Flexible Aged Care Programme and on specific supplements available for those living in remote communities [9]. Background Paper 2 of the Australian Royal Commission provides a very helpful summary of the impact of ageing on the need for aged care services, including trends in the aged care dependency ratio to 2050 and beyond [10]. The background paper notes how Aboriginal and Torres Strait Islander people tend to “have proportionally higher representation in home care services and proportionally lower representation in residential care services” [10]. The Australian Institute of Health and Welfare has estimated that health risk factors are more prevalent in outer regional and remote areas and that access to health care is more limited in these locations [11].

In an effort to fix decades-long issues of quality and safety in aged care provision, the Australian Government conducted the Royal Commission on Quality and Safety in Aged Care [12]. The report highlighted neglect, abuse, workforce, and funding issues, among many other problems. Geographical disparities in aged care services and inequity in access to aged care for older people living in regional, rural, and remote Australia were observed. This echoed findings expressed by the Royal Flying Doctor Service that “people living in regional, rural and remote areas …… have significantly less access to aged care than people living in major cities” [3] and that older people in rural and remote areas experience avoidable illnesses at higher rates. This is partly caused by the chronic skilled and workforce shortages outside metropolitan areas, which result in long waitlists, leaving many with no choice but to accept a much lower level of care [13].

This article examines the latest detailed publicly available national microdata to provide trends and estimates of the shortfall in aged care provision in rural and remote regions of Australia compared to metropolitan and urban areas, and to estimate how this will be exacerbated by the ageing of the population in those regions.

## 2. Materials and Methods

### 2.1. Study Design, Setting, and Participants

This observational study was based on a time series analysis of administrative data. We employed Australian government forecasts of population, and national survey data on health service utilisation in 2020 and 2021 to identify worsening gaps in aged care provision as the population of more remote communities ages. Data were accessed in February 2023, and we have developed estimates until 2032.

Aged care services are defined as ‘regulated care delivered in residential or community settings, including a person’s own home’ [14]. The majority of formal aged care in Australia is subsidised and supplemented by the Australian government and delivered by approved service providers. The Indigenous population aged 50 and over and the non-Indigenous population aged 65 and over are eligible to access aged care services through the MyAgedCare portal [15]. We included data from populations in all age ranges. Thus, some of the usage of aged care services in the younger age ranges reported in this article was by Aboriginal and Torres Strait Islander people, although there will be others living with disabilities in aged care facilities.

Ethics approval, consent to participate, and permission to access data were not required since the data were de-identified and publicly available.

### 2.2. Data Sources and Extraction

Two publicly available databases were accessed. Our primary source of population forecasts were those prepared by the Australian Bureau of Statistics (ABS) for the Australian Government Department of Health [16]. These provide forecasts for all states and territories, disaggregated by age and sex, at Statistical Area Level 2 (SA2) to 2032. Data on use of aged care services was obtained from the Australian Institute of Health and Welfare (AIHW) GEN database, a confidential database of 370,848 individuals who used residential, transitional, respite, and different levels of home care as of 30 June 2021 [17]. The latest release of these data was in April 2022; the population forecasts cover the period of 2021–2032. The previous release of the GEN data, with which we made some comparisons, comprised 335,889 individuals who used aged care services on 30 June 2020.

The data were assembled into Excel databases, with the primary unit of analysis being the government-defined Aged Care Planning Region (ACPR) [18]. We used a lookup table to relate location at Statistical Area Level 2 (SA2) codes to these ACPRs and then aggregated all entries for both males and females in each five year age range for both population and for various types of aged care facility use or home care package in the GEN database using Excel’s ‘SUMIF’ functions.

### 2.3. Data and Statistical Analysis

The Aged Care Planning Regions (ACPR) were classified according to geographical remoteness using the Modified Monash Model (MMM) scale [19], as depicted in Figure 1. The MMM is a key tool being increasingly used by the Australian Commonwealth Department of Health to “describe geographical access” [20]. In 2016, it was introduced into, for example, the Australian Longitudinal Study on Women’s Health (ALSWH) as a measure of remoteness [21].

Appendix A below provides Table A1 that lists the Aged Care Planning Regions by state and reports a simple average of the MMM scores for the Statistical Area 2 (SA2) areas that comprise them. In what follows, we group those ACPRs that have an average MMM score of 1.99 or below as ‘metropolitan and urban’, those from 2 and up to 5 as ‘regional and rural’ and 5 and above as ‘remote communities’.

Having aggregated for each ACPR, for males and females in each age range, the population and usage of permanent residential care, respite care, transition care, and home care packages at levels 1, 2, 3, and 4, we then computed the average usage of each per 1000 population. This was also computed for the three groupings of ACPRs noted above. We also calculated the amount of each type of aged care that would be used by those in remote communities if they used the same amount as those (in the same age range and gender) in metropolitan and urban ACPRs. We call the difference between this predicted amount and the amount of aged care actually used ‘shortfall’.

We then applied the population forecasts developed by the Australian Bureau of Statistics for the Australian Department of Health to the populations in each SA2 area and hence predicted the expected population of each remote community ACPR in the years 2025, 2028, and 2032. On the assumption that those in remote communities would receive aged care at the current level per 1000 population enjoyed by those in metropolitan and urban ACPRs, we computed the additional aged care facilities and packages that would be needed to meet their requirements. Finally, we added the shortfall in 2021 to the additional needs in each subsequent period to develop an estimate of the required additional service provision, by age group and gender, in each ACPR.

## 3. Results

In June 2021, those in the remote communities received a range of aged care services, as shown in Table 1. Overall, 12,419 individuals accessed aged care, of whom 7624 were in receipt of home care packages. Among the population aged over 65 (estimated from ABS data for each SA2 area as described above), some 8.4% enjoyed some kind of care service, rising to 35.7% for the group aged 85 and over.

Focusing on those aged 65 and over, Figure 2 shows that those in remote communities did indeed access less residential care per person than those in metropolitan or urban communities, that is, those in ACPRs with an MMM score of 1.99 or lower. 

As also noted above, we define ‘shortfall’ as the additional provision that would be required to bring the use of aged care services by those in remote community ACPRs to the same level per person as their counterparts of the same age and gender in metropolitan or urban ACPRs. In Table 2, we compute the additional provision that would be required to bring aged care service usage to an equivalent level across all age groups. An additional 1861 permanent residential and 159 respite and transition care places would need to be provided to bring provision for those aged 65 and over to the level enjoyed in metropolitan and urban ACPRs.

Partly to compensate for this shortage of residential places, Australia provides home care packages at a level that is assessed based on need. In 2020, there will be a shortfall in the provision of home care packages in remote communities. However, in 2021, at the peak of the COVID-19 crisis, the home care packages provided to those in remote communities were sometimes proportionately higher than those received by their urban and rural counterparts. Figure 3A,B highlight the difference between the two years: note that home care packages increased in all locations, but particularly so in remote locations.

The higher use of home care packages partly reflects the extra needs of Aboriginal and Torres Strait Islander people, which, of course, are more present in remote communities than in other ACPRs. The GEM data we are using to quantify aged care service use does seek to identify individuals as “Indigenous” or “non-Indigenous”. However, the recording of this data is imperfect, and a significant proportion of the individuals in the database are coded as “not stated/inadequately described”. Nevertheless, it is useful to understand the use of services by Indigenous people who have been identified as such.

Table 3 shows the use of aged care services by Indigenous people in all age ranges. As expected, there is significant use of both residential care and home care in the under-65 age groups. The dominant use of home care packages is at higher levels of care.

We then examined the impact of population ageing on the potential shortfall in aged care provision in remote communities for the total population. Figure 4 shows that the population is forecast to grow in all age groups. However, after 2030, the number of those aged 65–69 is expected to fall and that of those aged 80–84 to level off.

In what follows, we now focus on the care needs of those aged 65 and over. In addition to making up the shortfall for the groups aged 65 and over in 2021, it will be necessary to meet the needs of the ageing population. By 2025, there will be an additional 16,876 people aged 65 and over living in remote community ACPRs, of whom 5643 will be aged 80 and over. By 2032, this will have increased to 40,176 additional people aged 65 and over, of whom 19,190 will be aged 80 and over.

In Table 4, we bring together the population forecasts for both females and males with the data on penetration of aged care services to estimate the number of additional aged care services that will be needed both to make up the shortfall between provision in remote communities and that enjoyed by those in metropolitan and urban settings and to allow for expected population ageing. These figures are based on the population aged 65 and older. Additional services may also be required to meet the needs of Aboriginal and Torres Strait Islander people in younger age ranges, but that issue is outside the scope of the present study. We do not have population forecasts explicitly for the Aboriginal and Torres Strait Islander populations.

In addition to the 1861 permanent residential places, 159 respite and transition care places, and 543 home care packages that would be needed to bring aged care service levels to metropolitan and urban levels for those aged over 65, an additional 3222 residential places, 168 respite and transition care places, and 2986 home care packages will be needed to allow for population growth until 2032. Overall, we estimate that 8939 residential, respite, transition places, and home care packages will be needed to remedy the 2021 shortfall and meet the needs of the growing aged population until 2032.

Figure 5 below shows the extent of the shortfall in each of the remote community ACPRs. The bar charts are sized to provide an approximate indication of the additional total requirement to make up the 2021 shortfall and meet the needs of the ageing population up to 2032. The higher the bar, the greater the need. In a few ACPRs where the number of home care packages in 2021 is greater than needed to match urban and metro levels, no additional requirement for the 2021 shortfall is included in that category. A quick visual inspection of the map reveals that there are seven ACPRs with significant needs, including Wheatbelt, Great Southern, and Mid-West in Western Australia; Eyre Peninsula, Riverland, and South East in South Australia; and Orana Far West in New South Wales. The reasons for these large needs vary; for example, in ACPR 511, Wheatbelt, there is a large current shortfall in residential care places, whereas in ACPR 111, Orana Far West, the dominant reason is rapid population growth. A second factor that stands out from Figure 5 is that the largest difference between remote communities and their metropolitan or urban counterparts is a shortage of permanent residential aged care places.

## 4. Discussion

This article is relevant to Australian policy as it complements the existing knowledge of the aged care inequity experienced by older people in rural and remote Australia. Our findings highlight how the ageing population will exacerbate existing gaps in the provision of aged care in the next decade. Rural and remote communities will be hit the hardest. The current shortfall of 2000+ places in residential aged care will continue to widen. By 2032, to bring aged care service levels for those aged over 65 to similar levels to those enjoyed in metro-urban settings, a further 3000+ residential care places and 3000 home care packages will be required in remote communities alone.

Globally, contemporary issues affecting rural residential aged care provisions can be summarised into cognitive decline, aged care systems, workforce, admission and discharge, and end of life matters [22]. In contrast, our research demonstrates a geographical disparity and quantifies the magnitude of aged care provision within Australia’s rural and remote communities. Being a geographically large country with a small population presents its own challenges. The population is dispersed, and the lack of transport makes service provision difficult. As illustrated in Figure 5, while the shortfall in other ACPRs should not be neglected, a detailed investigation of how to address the problem in specific locations is warranted. Additionally, it is possible that people living in remote communities would prefer a different mix of aged care provision than those who live in urban communities. It may be that residential homes are more suited to families where relatives live in close proximity and wish to visit, for example, and that particular types of home care can be more cost-effectively provided in a metropolitan or urban setting. These are legitimate lines of inquiry, but we would argue that a starting point should be fairness and the equivalence of potential provisions. Deviations from this base should then only occur with the properly understood agreement of the communities concerned. Nevertheless, there are still shortfalls in all other categories; it is not the case that residential provision has been systematically replaced by home care packages in all locations.

The provision of residential care, transition and respite care, and home care packages for older people is only part of the story. Gardiner et al. [3] have set out how “Many Australians living in rural and remote areas of Australia need to travel hundreds of kilometres for health care services or to wait for health service providers, such as the Royal Flying Doctor Service (RFDS), to visit them”. They provide maps at Statistical Area Level 3, which show where there is no hospital or primary health clinic coverage within 60 min by motor vehicle. While healthcare and aged care are funded separately in Australia, there is a strong interface between the two systems. The funding model for aged care also disadvantages those where care provision involves travel: the cost of travel diminishes funds in the package for other care provision. The Aged Care Royal Commission identified chronic skilled workforce shortages, particularly outside the metropolitan area [2]. Our research shows thin markets occurring in rural areas, where often there is no provider available to deliver the care, resulting in the individual accepting a much lower level of care [13].

Addressing the shortage of health workers in rural, regional, and remote areas has been the focus of many government policy initiatives in recent years [23]. However, recent analysis of the availability of health professionals by remoteness still shows a much lower availability in terms of FTE per 100,000 population in geographic areas MMM 5, 6, and 7 than in metro, urban, and regional locations (MMM 1–4) [24]. Shortfalls in healthcare provision in remote regions are a major concern addressed in the Australian Government’s “Closing the Gap” programme between First Nations and non-Indigenous Australians. In the latest Closing the Gap Annual Report [25], a commitment is made to establishing an Australian Aged Care Commission, which will include a dedicated Aboriginal and Torres Strait Islander Commissioner responsible for “ensuring that appropriate aged care services are widely available for First Nations peoples”. The Australian government has dedicated a large amount of funding towards improving workforce shortages and developed new funding models and quality standards to tackle these issues for all population groups. Furthermore, the Australian Government has agreed with the Royal Commission’s recommendation to introduce a new Aged Care Act by 2023.

COVID-19 has further exacerbated the Australian aged care crisis. Pre-COVID-19, the lower level of the aged care workforce was predominantly occupied by a migrant workforce with poor working conditions and minimal training [26]. Ongoing workforce recruitment and retention challenges continue to exist, particularly outside metropolitan areas, even in the post-COVID-19 era [27]. Aside from addressing skilled workforce shortages to cater for complex care needs in later life, the shift in demographic profile, government policy, and funding demands more holistic planning [28]. While there is no single, short-term solution to address issues of this magnitude, the growing uptake of telehealth and technology may ease the delivery of skilled aged care provision. However, as the majority of aged care services, from residential aged care to home care packages, require local staff and face-to-face delivery, a multi-pronged approach is required beyond funding alone to be sustainable. Aligned with older people’s preference for ageing in place rather than living in residential care, our research shows that a whole-community approach to supporting their older population is needed when government-funded services are inadequate [29]. This may involve deploying local community members as volunteers to support older people and their caregivers to complement formal aged care provision.

A special issue of the *Australian Journal of Rural Health*, published in 2019, explored aspects of the impact of these issues on the quality of life for older Australians in rural and remote communities. Jackson et al. (2019), for example, provide guidance on how policy needs to be adjusted to ensure better service development, service access, and a sustainable workforce in such locations [30]. Research has shown how resilience can in fact lead to a surprisingly good quality of life for rural older people, even when aged care services are less available [31]. Resilience is also evident among some migrant aged care workforce, particularly those of Asian descent, who are able to draw upon individual and cultural values as well as support networks in the community [32]. Improving the interface between the wider health system and aged care has the potential to improve access to care, as is being explored in Western Australia [33].

There are strengths and limitations to this paper. We analysed publicly accessible administrative data to quantify and locate the current and projected gaps in aged care provision in small, rural, and remote communities in Australia. As a result of the Royal Commission into Aged Care Quality and Safety, the Australian government has put in place several of its key recommendations. Nearly two-thirds of residential aged care providers are operating at a deficit [34], and this is partly attributed to a low level of occupancy. Residential aged care providers outside metropolitan areas are heavily affected by increased regulatory compliance, a workforce shortage, and wage pressure and may move away from the provision of residential care, which will further exacerbate the undersupply of this form of care. The impact of these various changes is difficult to predict, but it is likely that the widening inequity for older people in rural and remote communities will persist. Further studies are needed to address the worsening inequity in accessing aged care and to explore a sustainable model of aged care for older people living in small rural and remote communities to age in place.

## 5. Conclusions

We have provided a snapshot of the future of Australian aged care provision in remote communities by 2032 to inform policy and practise. In working out how to respond to the shortfalls identified in this article, sensitive optimisation of all available methods of improving the quality of life of the ageing population in rural and remote communities is essential. As the ageing population is growing and, with it, more complex care needs, and in light of ongoing aged care reform and workforce shortages, it is necessary to review the existing aged care model in order to deal with the shortfalls identified in this article.

## Figures and Tables

**Figure 1 geriatrics-08-00047-f001:**
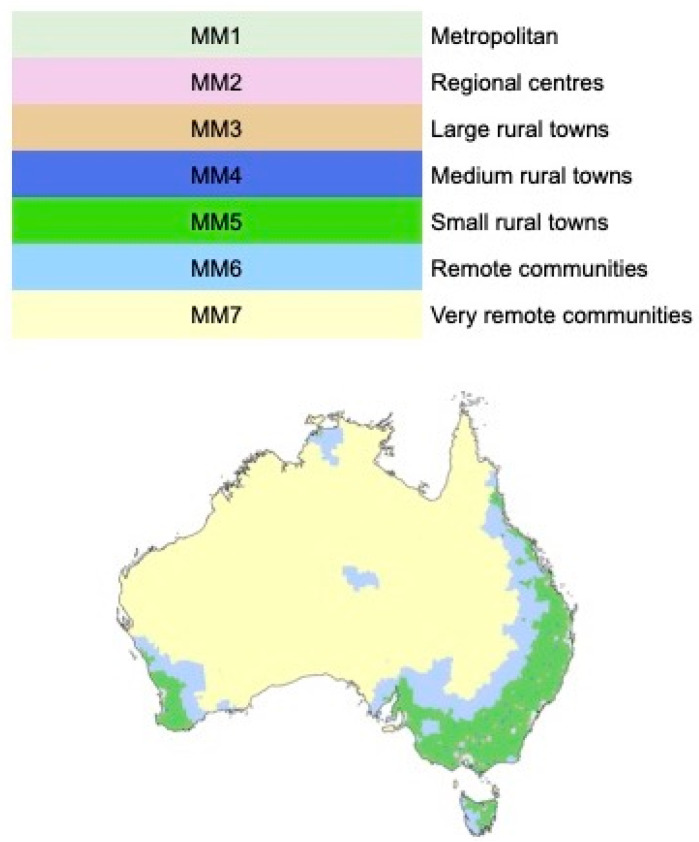
Modified Monash Model (MMM) scale for classification of geographical remoteness.

**Figure 2 geriatrics-08-00047-f002:**
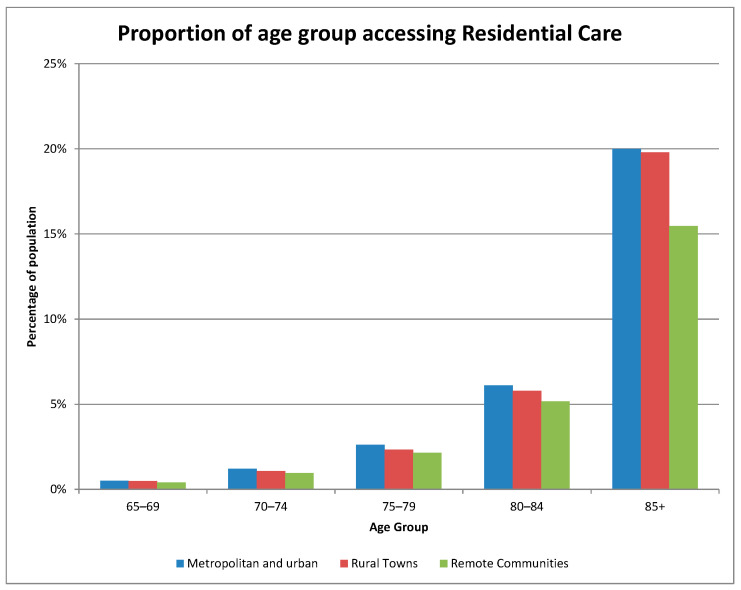
Access to residential aged care services in remote communities in 2021.

**Figure 3 geriatrics-08-00047-f003:**
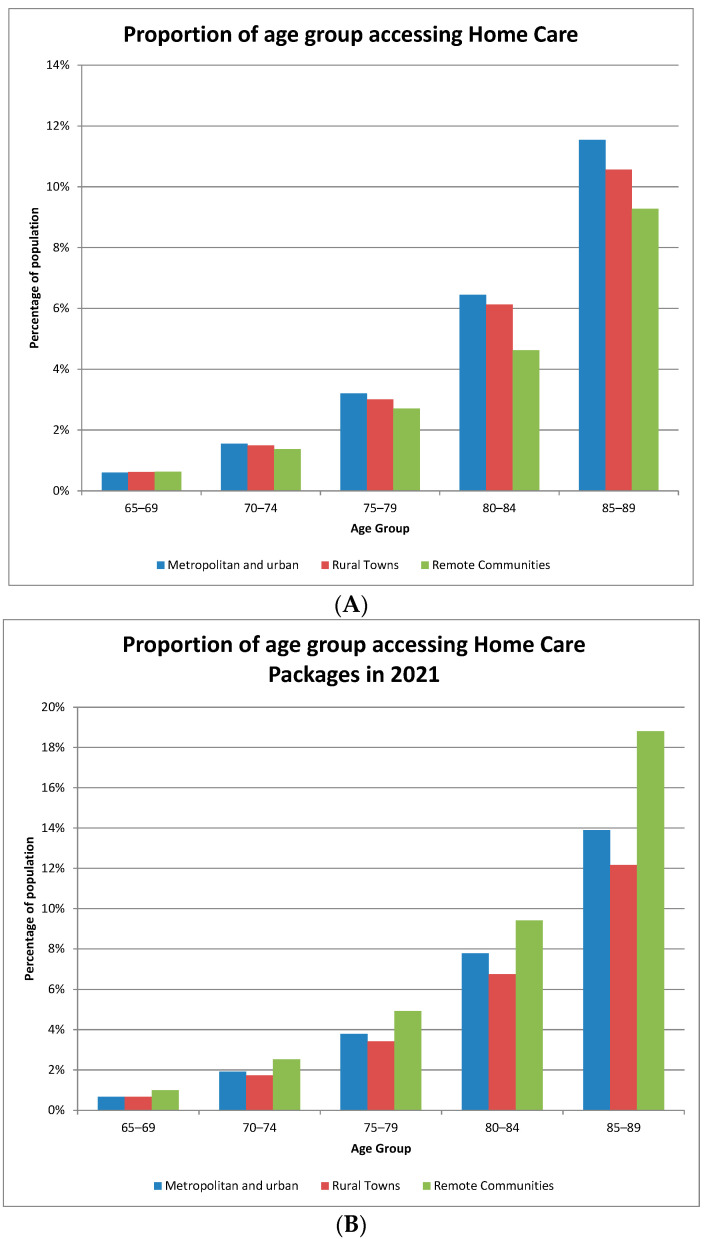
Access to home care packages in remote communities in 2020 and 2021. (**A**): Access to home care packages in remote communities in 2020; (**B**): Access to home care packages in remote communities in 2021.

**Figure 4 geriatrics-08-00047-f004:**
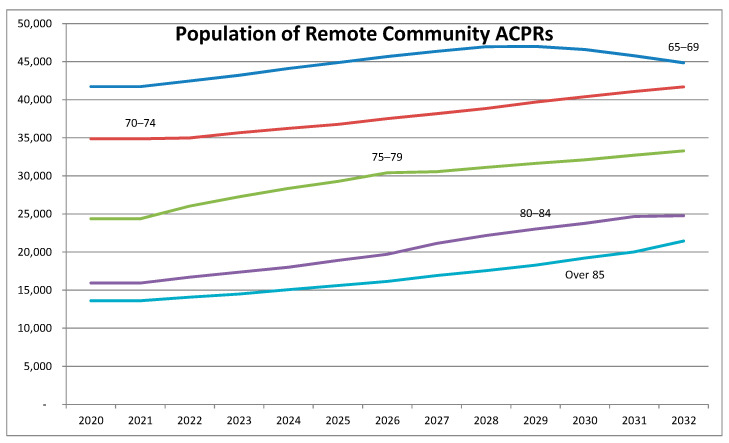
Growth in population of remote community ACPRs.

**Figure 5 geriatrics-08-00047-f005:**
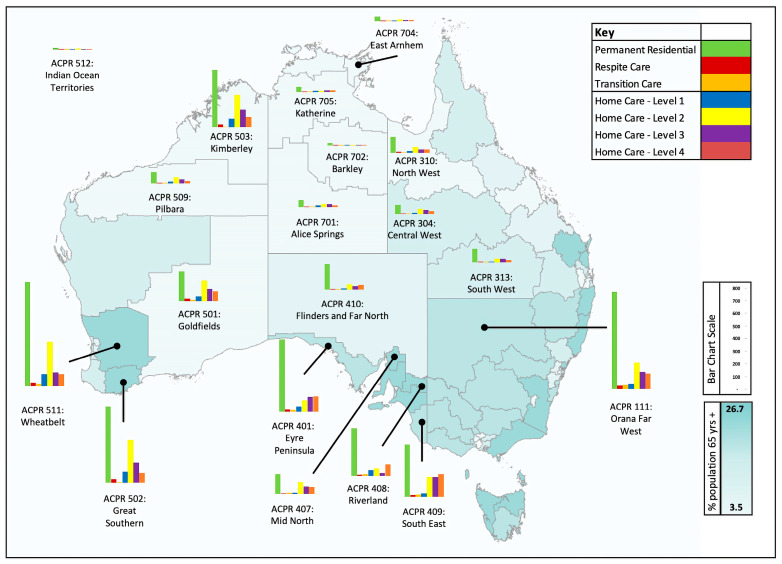
Location of the aged care shortfall in remote communities. Based on Public Health Information Development Unit (PHIDU) and Torrens University Australia material from: *Social Health Atlas of Older People in Australia*, Aged Care Planning Regions (ACPR). Published 2021. [https://phidu.torrens.edu.au/current/maps/sha-topics/ageing/acpr-single-map/atlas.html (accessed on 21 April 2023)].

**Table 1 geriatrics-08-00047-t001:** Use of aged care services in remote community ACPRs in June 2021.

	Age Ranges									
Care Type	0–49	50–54	55–59	60–64	65–69	70–74	75–79	80–84	85+	All
Permanent Residential	7	29	35	94	189	378	594	928	2344	4598
Respite Care	1	1	1	5	7	16	29	41	70	171
Transition Care	-	-	-	-	1	3	3	8	11	171
Home care—Level 1	1	2	9	12	66	122	236	292	427	1167
Home care—Level 2	-	7	39	66	179	384	528	718	1,268	3189
Home care—Level 3	-	3	21	33	134	269	320	363	692	1835
Home care—Level 4	1	5	24	29	94	228	272	315	465	1433
Any Service	10	47	129	239	670	1400	1982	2665	5277	12,419

**Table 2 geriatrics-08-00047-t002:** ‘Shortfall’ in aged care services for remote community ACPRs in 2021.

	Age Ranges									
Care Type	0–49	50–54	55–59	60–64	65–69	70–74	75–79	80–84	85+	All
Permanent Residential	(1)	(18)	(6)	(18)	22	98	186	327	1229	1818
Respite Care	(1)	(1)	(1)	(4)	4	1	3	11	44	56
Transition Care	-	0	1	1	8	14	22	23	29	99
Home care—Level 1	(1)	(1)	(7)	(10)	(8)	(14)	(90)	(121)	(289)	(541)
Home care—Level 2	0	(4)	(30)	(50)	(42)	(14)	7	6	(379)	(504)
Home care—Level 3	1	1	(13)	(18)	(31)	(47)	6	72	(62)	(91)
Home care—Level 4	0	(1)	(14)	(10)	(5)	6	16	40	150	182
Any Service	(1)	(24)	(70)	(108)	(52)	45	149	359	722	1019

Notes: The numbers in brackets reflect situations where remote communities have higher use of the relevant care service than age equivalent people elsewhere. The numbers in brackets in the ‘All’ column show that one effect of the shortage of residential places is a higher use of home care packages.

**Table 3 geriatrics-08-00047-t003:** Use of aged care services by individuals identified as Indigenous in 2021 in remote community ACPRs.

	Age Ranges									
Care Type	0–49	50–54	55–59	60–64	65–69	70–74	75–79	80–84	85+	All
Permanent Residential	6	16	19	45	46	60	56	56	78	382
Respite Care	1	1	-	4	1	5	3	3	3	21
Transition Care	-	-	-	-	-	-	-	-	-	-
Home care—Level 1	-	1	6	7	9	5	8	3	1	40
Home care—Level 2	-	-	20	37	55	51	42	20	28	253
Home care—Level 3	-	1	12	10	17	28	14	7	6	95
Home care—Level 4	-	1	11	19	20	16	17	17	14	115
Any Service	7	20	68	122	148	165	140	106	130	906

**Table 4 geriatrics-08-00047-t004:** Summary of additional aged care services needed to make up the shortfall and allow for population ageing.

Aged Care Type	Remedy 2021 Shortfall	Growth 2022–2025	Growth 2026–2028	Growth 2031–2032	Total
**Permanent Residential**	1861	976	890	1356	5084
**Respite Care**	62	35	32	46	175
**Transition Care**	97	19	16	20	152
**Home Care—Level 1**		93	76	88	257
**Home Care—Level 2**	202	410	352	454	1418
**Home Care—Level 3**	119	263	227	300	909
**Home Care—Level 4**	222	238	205	280	945
**Total**	**2564**	**2034**	**1797**	**2544**	**8939**

## Data Availability

The primary datasets used for this analysis were: population forecasts developed by the Australian Bureau of Statistics for the Australian Department of Health for the populations in each SA2 area, available at https://www.gen-agedcaredata.gov.au/Resources/Access-data/2019/September/Population-projections,-2017-(base)-to-2032-for-al. (accessed on 20 October 2021); Australian Institute of Health and Welfare (AIHW) GEN database, for both 30 June 2020 and 30 June 2021, available at https://www.gen-agedcaredata.gov.au/Resources/Access-data/2022/July/GEN-data-Admissions-into-aged-care (accessed on 16 February 2023).

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
