# Peer review of "The Impact of Population Ageing on Rural Aged Care Needs in Australia: Identifying Projected Gaps in Service Provision by 2032"

_geriatrics, 2023, doi:10.3390/geriatrics8030047_

Round 1

Reviewer 1 Report

This is a well-resourced, comprehensive and critical analysis of aged care service provision across Australia, presenting sound evidence of reasons for disparities that exist in rural and remote regions. 

There is one statement that i feel needs clarification - page 10, last sentence, line 308. Explain the relationship between the resilience of rural-based older people and the resilience of some migrant aged care workforce members. These appear to be two separate issues, so please explain how one supports the other. 

Author Response

Thank you for your supportive comments and suggestions.

We have added explanation to the sentence on p10. The links to tables and figures have been fixed. We have also made an improved version of Figure 5.

Reviewer 2 Report

This is an interesting paper highlighting the geographical variation - disparity in the provision of a range of aged care services in Australia with a focus on remote areas of the country - it essentially highlights what has been the pattern for some time now and projects forward the expected shortfall by area with the ageing of the population - it quantifies it through publicly available data.

There are a number of minor changes to the paper that could improve its value to the reader:

The introduction does not mention (review) previous studies that have looked at spatial distribution of aged care services in Australia and in other akin fields like health services and disability services particularly as the authors talk about the interconnectivity of these service areas.

One would suspect that the government (Dept of Health) has undertaken a detailed study of the disparities by area but this research is probably not available to the public.

Figure 1 and Figure 2, Figure 3  are too small to distinguish the regions by colour and read the axes respectively.

Table 2 should have a footnote indicating that the numbers in brackets are an oversupply in remote communities. Another footnote should explain the disparity in total numbers for each category of aged care stated in text (lines 145,146, 165) as opposed to when the data is broken down by age in Table 2.

The placement of Lines 160-167 are confusing as this discusses the need to counter the shortfall in remote areas after just discussing and displaying the increase in home care packages in remote areas compared to metro-urban areas. The reason for this is well outlined in the next paragraph so these two paragraphs just need re-organising.

Line 176-77 would benefit by just adding to the end of the sentence with regard to Indigeneity ‘however at times this is recorded’.

Table 3 ‘indigenous’ = ‘Indigenous’

Line 187-188 this sentence should end with ‘for the total population.’ As we have just read about the Indigenous population it is initially confusing if the paragraph is still discussing this population or the overall total population.

Table 4 Table headings ‘Growth in over-65’s population’ = ‘Growth in over-65s population

Line 222 ‘particularly large needs’ would be better if expressed as ‘significant needs’

Line 238 ‘places in residential aged care places will’ = ‘places in residential aged care will’

Line 240 ‘in urban/rural settings’ should this be metro-urban settings as through rest of paper don’t talk about urban/rural settings

Lines 246, 247, 248 yes these are important points but it is also the costing model for aged care – cost of travel diminishes funds in package for actual services

Line 252-254 need some references for these thoughts particularly around the role of families in providing care – once an older person enters residential care people including families do not visit despite proximity.

Lines 282-284 here are you referring just to needs of Indigenous population or for all population groups?

Line 287 – need clarity around pre and post covid comments – is it being suggested that the migrant workforce is no longer available due to covid?

Line 298-299 ‘our research shows the importance of whole of the rural community to support their older population’. The research does not show this. The conclusion that can be drawn from the study is that in the absence of service provision through government funded services a whole of rural community approach is needed.

Lines 311-315 this is not a limitation of this study – the results of the reforms to the aged care system cannot be, at this point in time, anticipated by anyone.

Line 315-317 just need to extend this statement to say that many may move away from the provision of residential care which will exacerbate the undersupply of this form of care.

The final point to consider is that the paper in the introduction mentions the interface between health and aged care but then doesn’t continue that discussion in terms of how that interface might work to improve access to care – see for example Government of WA, WA Country Health Service, WA Country Health Service Strategy for Older People 2022-2027.

Author Response

Thank you for these very helpful comments, which we agree improve the article.

We have made reference to AIHW studies that highlight the spatial difference of health services, and also highlighted where references we already cite include locational research.

The size of Figure 1, 2 and 3 has been increased

Footnotes have been added to Table 2 as suggested

Indigenous has been capitalised in Table 3

"For the total population" added in Lines 187-188

Over-65s heading problem changed

Repeated word 'places' removed in Line 238

Have added a sentence about cost of travel reducing funds for other purposes

urban/rural changed to metro-urban in line 240

Added 'and wish to visit' line 252

Clarified that it is whole population in lines 282-284

Added phrase to distinguish between pre- and post-COVID responses

Adjusted lines 298-299 as suggested

Clarified argument about limitations in lines 311-317

Added citation and reference to WA Country Health Service initiatives. Thank you for bringing this to our attention.

Reviewer 3 Report

Comments are on the pdf attached

Author Response

Thank you for these helpful comments. In response we have:

Removed the word ‘small’ when describing rural areas

Noted that Aboriginal and Torres Strait Islander people are eligible for aged care from age 50 onwards

Clarified that by Annex we meant Appendix A

Links for references to figures and tables have been fixed

Added explanation to refer back to previous paragraph about source of population estimates, line 138

Changed ‘outstripped’ to ‘were proportionally higher’ in line 152

Tightened language and headings to make clear we are talking about Home Care Packages in Figure 3. Thank you for spotting this.

Indigenous capitalized

Simplified Table 4 and adjusted text to make more understandable. Thank you for this suggestion.

Have adjusted text to make clear what we are saying about resilience in migrant communities

Adjusted the concluding paragraph

Reviewer 4 Report

In general, the article is well-written, relevant and brings interesting information to discussion in a clear way. The scientific review for the background is not sound but it is not the purpose of the paper, thus it is acceptable as it is. The methods are explained in a very simplistic manner but also acceptable given the aim and the methods used.

As suggestions for change:

- The links to figures display an error, thus it ws sometimes difficult to follow the arguments / understand whch table or figure is referred (e.g. lines 104, 136, 141, etc) but that can be easily solved in a revision.

- Figure 5 is not displyed in enough quality to be readable.

- Propose to replace this statement with something less final "This article is unique as it complements the..." (line 234), by adding, e.g. it is unique to the extent of our knowledge, or replacing the word unique by relevant, for example.

Author Response

Thank you for your helpful comments and suggestions.

The links to tables and figures have been fixed. We have also made an improved version of Figure 5.